# Novel and Conventional Isolation Techniques to Obtain Planctomycetes from Marine Environments

**DOI:** 10.3390/microorganisms9102078

**Published:** 2021-10-01

**Authors:** Inês Vitorino, José Diogo Neves Santos, Ofélia Godinho, Francisca Vicente, Vítor Vasconcelos, Olga Maria Lage

**Affiliations:** 1Departamento de Biologia, Faculdade de Ciências, Universidade do Porto, 4169-007 Porto, Portugal; zesantox@gmail.com (J.D.N.S.); ofeliagodinho95@gmail.com (O.G.); vmvascon@fc.up.pt (V.V.); olga.lage@fc.up.pt (O.M.L.); 2CIMAR/CIIMAR, Centro Interdisciplinar de Investigação Marinha e Ambiental, Universidade do Porto, 4450-208 Matosinhos, Portugal; 3Fundación MEDINA, Centro de Excelencia en Investigación de Medicamentos Innovadores en Andalucía, 18016 Granada, Spain; francisca.vicente@medinaandalucia.es

**Keywords:** *Planctomycetia*, iChip, ecology, macroalgae, sediments, invertebrates, novel taxa

## Abstract

Bacteria from the distinctive *Planctomycetes* phylum are well spread around the globe; they are capable of colonizing many habitats, including marine, freshwater, terrestrial, and even extreme habitats such as hydrothermal vents and hot springs. They can also be found living in association with other organisms, such as macroalgae, plants, and invertebrates. While ubiquitous, only a small fraction of the known diversity includes axenic cultures. In this study, we aimed to apply conventional techniques to isolate, in diverse culture media, planctomycetes from two beaches of the Portuguese north-coast by using sediments, red, green, and brown macroalgae, the shell of the mussel *Mytilus edulis*, an anemone belonging to the species *Actinia equina,* and seawater as sources. With this approach, thirty-seven isolates closely related to seven species from the families *Planctomycetaceae* and *Pirellulaceae* (class *Planctomycetia*) were brought into pure culture. Moreover, we applied an iChip inspired in-situ culturing technique to successfully retrieve planctomycetes from marine sediments, which resulted in the isolation of three additional strains, two affiliated to the species *Novipirellula caenicola* and one to a putative novel *Rubinisphaera*. This work enlarges the number of isolated planctomycetal strains and shows the adequacy of a novel methodology for planctomycetes isolation.

## 1. Introduction

Planctomycetes are bacteria capable of colonizing a wide range of habitats. They display diverse metabolic capacities and play major roles in the nitrogen and carbon cycles [1,2]. Presently, aquatic environments exhibit the most known diversity in this phylum, with a vast predominance in marine environments [1]. Planctomycetes are associated with organisms such as macroalgae, plants, and invertebrates [3,4,5,6,7,8,9,10,11]. They are also found in diverse terrestrial environments such as peat bogs [12,13,14]. The great capacity for adaptation of the Planctomycetes is also supported by their ability to colonize even extreme habitats such as hydrothermal vents, hot springs, deep-surface locations, and other anaerobic environments [15,16,17,18,19,20]. Taxonomically, the phylum *Planctomycetes* is divided into the classes *Phycisphaerae* and *Planctomycetia* [21,22]. Additionally, the *Candidatus* order "Brocadiales" from the *Candidatus* class “Brocadiae” is constituted by the anaerobic ammonium oxidation (anammox) planctomycetes, which do not have axenic cultures available as of now [23].

Planctomycetes display diverse respiration and cell division modes. While *Phycisphaerae* members are mostly anaerobic and divided by binary fission, the common bacterial cell division mode, *Planctomycetia* members are mostly aerobic and divide by budding [1,2,24,25]. They have unique features that stand out compared to other bacteria, such as (1) complex invagination systems of the cytoplasmic membrane, (2) occasionally large periplasms (3) division without the universal division protein FtsZ (the associated mechanism still not determined), (4) rare carotenoids, (5) phagocytosis-like cell engulfment (which was thought to be exclusive to eukaryotes) and (6) anaerobic ammonium oxidization by the anammox planctomycetes that possess (7) a compartmentalized cell structure named the anammoxosome [1,26,27,28,29]. Planctomycetes can have large genomes, but only about half of the genes have predicted function. Furthermore, their genomes show a considerable number of biosynthetic gene clusters, which are often related to the production of secondary metabolites [1,30,31]. In fact, they were shown to be biotechnologically interesting bacteria, as they can produce antimicrobial and anti-cancer compounds and can be used as a supplementary food source (single-cell pigment or single-cell protein), as demonstrated in the model organism *Daphnia magna* [30,32,33,34,35,36,37]. Thus, planctomycetes are bacteria with great potential in diverse fields.

Although Planctomycetes are present in many environments, the diversity of isolated strains is currently very low compared to the diversity uncovered from metagenomics and environmental studies [1]. Planctomycetes can be quite abundant in environmental samples but their isolation into pure culture is not always a straightforward process; they are slow growing bacteria with high generation times and they can be easily overtaken by fast growing bacteria [2]. Aside from several successful isolation attempts, including those of Dr. Heinz Schlesner in the 20th century, few representatives of this phylum were available in pure cultures [38]. However, in the last two decades, several diverse improvements into the targeted isolation of planctomycetes were developed, which lead to the isolation and study of a considerable number of planctomycetal strains and the discovery of new taxa [7,24,39,40,41]. Thus, it is essential to continue to unveil the taxonomic diversity of planctomycetes and their unique biology.

Based on the fact that most of the existing microbial diversity remains unculturable which is mainly due to the inadequacy of artificial media to enable growth, diverse in-situ culturing systems have been developed in the last few years to help overcome this problem [42,43,44]. One of these systems is the isolation chip (iChip) [42]. This device contains a plate with multiple wells that are inoculated with a suspension of environmental bacterial cells, gelatinized or not, and then covered with membranes and sealed, turning each well into a miniature diffusion chamber [42]. The device is then incubated in-situ (in the cell’s original environment), enabling the bacteria to grow with natural nutrients and growth factors from their original habitat by diffusion [42]. This methodology has proven to increase colony counts as well as the isolation of phylogenetic microbial uniqueness [42,45]. In the original iChip culturing system, it was intended to have an average of just one cell per mini diffusion chamber through previous dilutions of the sample, which allowed for an immediate obtainment of pure cultures [42]. A recent adaptation of this technique, designated FACS-iChip, combines flow cytometry-based fluorescence-activated cell sorting with an adapted iChip structure. This technique proved to guarantee the sorting of a unique cell per well [46]. Furthermore, the iChip is a high throughput system with access to a great diversity of microorganisms, and a high probability to find innovative bioactive compounds [47,48]. In fact, teixobactin was the first compound of a novel class of antibiotics with potent activity to be isolated from a previously undescribed soil microorganism retrieved with the iChip [49,50]. 

In this work, we aimed to assess the diversity of planctomycetes associated with sediments, red, green, and brown macroalgae, the anemone *Actinia equina*, the shell of the mussel *Mytilus edulis,* and the surrounding seawater of these organisms. Materials were collected from two beaches on the Portuguese north-coast using conventional isolation techniques. We also experimented with different media formulations with the objective of selecting and enhancing planctomycetes growth. Furthermore, a novel technique for the isolation of new planctomycetes from marine sediments through the adaptation of the in-situ iChip culturing system was successfully conducted.

## 2. Materials and Methods

### 2.1. Sampling and Isolation 

Three independent isolation experiments focused on obtaining planctomycetes from marine habitats, covering different locations, time periods, diverse sources, methodologies, and medium formulations (Table 1 and Table 2).

The first isolation study (Table 1) was focused on the use of conventional planctomycetes isolation techniques. Biological material was collected in October of 2018 from a rocky beach located in the Portuguese north-coast (Luz beach in Porto: 41°09′12.38 N, 8°40′45.91 W), which included thalli from the green macroalga *Ulva* sp., the red algae *Porphyra dioica* and *Chondrus crispus,* the brown alga *Fucus* sp., and sea water. Samples were picked during low tide; the temperature of the water was around 10 °C and the samples were immediately brought in a cold chamber to the laboratory. Pieces of the thallus of each macroalga (approximately 1.5 × 1.5 cm) were washed in sterile sea water to remove non-attached bacteria, containing 1% econazole nitrate (Pevaryl®), an anti-fungal agent used to prevent posterior fungal contamination. Pieces of the algae were then placed directly onto agar plates of different media: modified M13 medium (M13), 1:10 M13 medium, and ammonium sulfate medium (ASM) [7]. All media were supplemented with 1% econazole nitrate and with or without ampicillin (200 mg/L) and streptomycin (1000 mg/L) (Table 2). The antibiotic mixture and the anti-fungal solution were used to prevent bacterial growth of non-target bacterial and fungal contaminations, respectively.

The biofilm of the algae thalli was also scrapped, and the obtained extracts suspended in 1mL of sterilized sea water and 100 µL inoculated onto agar plates of the referred media. Furthermore, 500 mL of the collected sea water was filtered through a 0.22 µm pore filter, the obtained cells were then re-suspended in sterile sea water and 100 µL of this suspension was inoculated onto the referred media. All cultures were incubated at 25 °C until visible growth was evident; it was then transferred to a new medium until pure cultures were obtained. Colonies with common planctomycetal pigmentation and other features were selected based on cell morphological characteristics, such as cell aggregation in rosettes, and cell division by polar budding and/or pear-to-spherical cell shape, which are traits indicative of the presence of planctomycetes. Pure cultures were then maintained in their respective isolation medium and cryopreserved at −80 °C, in 20% (*v*/*v*) glycerol.

The second isolation experiment (Table 1), which also relied on the use of traditional techniques utilized for the isolation of planctomycetes, was performed in a different location on the Portuguese north coast, the Memória beach in Matosinhos (41°13′50.35″ N 8°43′16.61″ W), in March of 2020, and focused on exploring different sources such as sediments, sea water, and the invertebrates *Mytilus edulis* and *Actinia equina*. Approximately six mussels from the species *Mytilus edulis* were collected during low tide, with the temperature of the water around 12 °C, and immediately brought into the laboratory in a cold chamber. Biofilm from the outside of the shell of all mussels was scrapped, re-suspended in 100 µL of sterilized sea water, and inoculated on modified M14 and M13+NAG medium plates (Table 2) supplemented with 1% (*w*/*v*) econazole nitrate and the following antibiotics: vancomycin (4 µg/mL) or ampicillin (200 mg/L) and streptomycin (1000 mg/L), or imipenem (2 μg/mL) or ciprofloxacin (2 μg/mL). One specimen of the anemone *Actinia equina* was collected and macerated with 5 mL of sterile sea water, and 100 µL was inoculated on the aforementioned media. Seawater was collected in sterile Schott flasks, from which 1 L was filtered through a 0.22 µm pore filter, re-suspended in 10 mL of sterile seawater, and 100 μL was inoculated in each agar plate of the aforementioned media. The inoculums were incubated at 25 °C until growth was visible. One gram of the collected sand was added to sterile falcon tubes with 25 mL of the previously mentioned broth media and incubated for enrichment at 25 °C, with 220 rotations per minute (r.p.m.), for approximately 12 months. An aliquot of 100 μL was then spread on the respective agar medium. The selection of planctomycetal-like colonies was performed in the same way as previously mentioned. The isolates obtained were cryopreserved in medium supplemented with 20% (*v*/*v*) glycerol, at −80 °C.

The third isolation experiment (Table 1) was also performed at Memória beach in October 2020, and it was based on both conventional and novel techniques for the isolation of Planctomycetes. During low tide, with water temperatures of around 12 °C, the red algae *Corallina* sp. and *Porphyra dioica*, the green algae *Ulva* sp. and *Codium* sp., the mussel *Mytilus edulis*, the anemone *Actinia equina*, and seawater and water submerged sediments were collected and immediately brought into the laboratory in a cold chamber. Macroalgae thalli were treated similarly as described in the protocol for the first isolation experiment. Pieces of the thalli of each macroalgae (approximately 2 × 2 cm) were washed in sterile sea water containing cycloheximide (20 mg/L) (another anti-fungal agent to prevent posterior fungal contamination) and then placed on agar plates of modified M13 medium and on N-acetylglucosamine medium (NAGM) (Table 2), both supplemented with cycloheximide (20 mg/L), ampicillin (200 mg/L), and streptomycin (500 mg/L) [7]. The biofilm of each macroalga was scrapped, and the obtained extracts were suspended in 1 mL of sterile sea water, from which 100 µL was inoculated onto agar plates of the same media. The invertebrates (mussels *Mytilus edulis* and anemone *Actinia equina*) were treated similarly to the ones from the second isolation experiment. The biofilm from the outside shell of six specimens was scrapped, re-suspended in sterile sea water, and 100 µL of this suspension was inoculated on agar plates of modified M13 medium and NAG medium (NAGM), both supplemented with cycloheximide (20 mg/L), ampicillin (200 mg/L), and streptomycin (500 mg/L) (Table 2) [7]. One specimen from the sea anemone was macerated with 5mL of sterile sea water, and 100 µL was inoculated onto the same medium agar plates. Half a liter of the collected sea water was filtered through a 0.22 µm pore filter; the filter was then placed directly on agar plates of modified M13 medium and on NAG medium, both supplemented with cycloheximide (20 mg/L), ampicillin (200 mg/L), and streptomycin (500 mg/L) (Table 2) [7]. All inoculums were incubated at 25 °C until growth was visible, and the selection of planctomycetal-like colonies was based on the observation of characteristics traits already mentioned. The isolated strains were cryopreserved in a medium supplemented with 20% (*v*/*v*) glycerol, at −80 °C.

A novel methodology was developed to isolate Planctomycetes from marine sediments (Table 1). The isolation system utilized was based in the iChip culturing system and one 96-Well Filtration Plate MultiScreen® was used to substitute the assembled structure originally described (scheme briefly summarized in Figure 1) [42]. A bacterial inoculum was obtained after the vigorous shaking of a mixture of 25 g of wet sediments (collected in October 2020) in 10 mL of sterile sea water. Each well of the plate was filled with 100 µL of the bacterial inoculum 10× diluted in agarized sea water (0.3% agar, *w*/*v*) supplemented with cycloheximide (20 mg/L), ampicillin (200 mg/L), and streptomycin (500 mg/L). The plate was then sealed with parafilm^®^ on the upper part and placed with the filter side down in between layers of slightly humid sediments (collected from the same sampling location) in a container. After approximately 45 days of incubation at room temperature (±20 °C) in the darkness, the content of each well was re-inoculated onto modified M13 medium plates and incubated at 25 °C until colonies appeared [7]. A standard enrichment technique was also performed: 25 g of sediments was inoculated into 25 mL of M13+NAG liquid medium (Table 2). After incubation for 45 days, at 25 °C and 200 r.p.m., 100 µL was inoculated onto modified M13 plates. Pure cultures with traits indicative of the presence of planctomycetes were cryopreserved at −80 °C, in medium supplemented with 20% (*v*/*v*) glycerol. 

### 2.2. Phylogenetic Inference of Isolates

The genomic DNA of the isolates was obtained using the extraction kit E.Z.N.A. Bacterial DNA Isolation Kit (Omega BIO-TEK Norcross, GA, USA). For the identification of isolates, an analysis of the 16S rRNA gene was carried out. The gene was amplified with the primers 27F and 1492R, the PCR performed as previously described by Bondoso et al., and the PCR products purified using the Illustra™ GFX™ PCR DNA and Gel Band Purification Kit [51]. Sequencing was performed at Eurofins Genomics. The analysis of sequences was carried out using Geneious R11 software. The affiliation of the isolates was determined by the analysis of the partial 16S rRNA gene sequences obtained using the 16S-based ID tool in the EZBioCloud platform [52]. The known thresholds for taxonomic inference (<98.7 for novel species and <94.5 for novel genus) were used [53]. The 16S rRNA gene sequences of the isolates and type species of close relatives (retrieved from the National Center for Biotechnology Information (NCBI) database) were aligned using CLUSTAL W and the phylogenetic tree computed using MEGA7 software with the Maximum Likelihood method, 1000 bootstraps replicates, the General Time Reversible mode, and with the Gamma distributed with Invariant Sites (G+I) option [54,55]. All 16S rRNA gene sequences of the isolates were deposited in the NCBI database under the GenBank accession numbers MW669943-MW669975, MZ687805-MZ687809, and MW588631. Furthermore, for comparison of the ecology of the isolated strains with the described taxa, additional information on other isolates belonging to the related described species was obtained by searching for hits with >98.7% similarity of the 16S rRNA gene in the NCBI database with the BLAST search. 

## 3. Results and Discussion

In this study, we sampled two beaches from the Portuguese north-coast, which, although not located many kilometers apart from each other, present different characteristics: Memória beach is located in Matosinhos, and its sand extension is higher than the one in Luz beach, which is smaller, composed of rockier areas, and located right in the heart of the residential area of Porto, with greater human pressure and increased pollution. They also presented different macroalgae and invertebrate communities during the sampling periods. With our approach to explore these two different locations, culture medium formulations (Table 2) and isolation sources and techniques (Table 1), we successfully brought into pure culture a total of forty novel planctomycetal isolates. Figure 2 shows the diversity obtained from each source, and in Figure 3, a 16S rRNA gene based phylogenetic tree shows the relationships of this diversity within the phylum. Table 3 summarizes strains designation, details of the isolation methodology, and their phylogenetic affiliation. The analysis of the 16S rRNA gene with the species cut-off of 98.7% showed that the isolates obtained are related to eight different planctomycetal species, *Rhodopirellula baltica*, *Rhodopirellula lusitana*, *Novipirellula caenicola*, *Novipirellula rosea*, *Rubinisphaera brasiliensis*, a putative novel *Rubinisphaera*, *Gimesia chilikensis*, and *Alienimonas chondri* [53]. These species belong to the Class *Planctomycetia*, and specifically to the families *Pirellulaceae* (*R. baltica, R. lusitana, N. rosea*, and *N. caenicola*) and *Planctomycetaceae* (*A. chondri*, *G. chilikensis*, and *Rubinisphaera* genus) [56].

### 3.1. Macroalgae as Source for Planctomycetes

The majority of the planctomycetal isolates obtained in this study originated from macroalgae (Figure 2). In October 2018, *Rhodopirellula baltica* dominated the Planctomycetes isolated from the community of the macroalgae tested (Figure 2 and Table 3), especially from *Ulva* sp. (seventeen isolates) and *Porphyra dioica* (four isolates). Only one strain was retrieved from *Fucus* sp., which is affiliated with *Rhodopirellula lusitana,* and two strains from *Chondrus crispus*, LzC1 and LzC2^T^, which belong to a novel taxon that was recently described based on a polyphasic description study, *Alienimonas chondri* [57]. In October 2020, one strain closely related to *Novipirellula caenicola* was retrieved from *Ulva* sp., and another strain affiliated with *R. baltica* from *Corallina* sp. No strains were obtained from the other macroalgae tested (*Codium* sp. or *Porphyra dioica*). In these two isolation experiments, both thalli portions and the scrapped biofilm of each macroalgal surface were used, and almost all strains obtained originated from the portions (isolates from *Ulva* sp. and *Porphyra dioica* and *Fucus* sp.) rather than from the surface biofilm (isolates from *Chondrus crispus* and *Corallina* sp.). In fact, it was previously demonstrated that planctomycetes can be strongly associated to macroalgae communities, which is consistent with the results here obtained [6]. The majority of the isolates from October 2018 (91.7%) were obtained in the more oligotrophic media 1:10 M13 and ASM, while only three strains were isolated in M13 medium. In October 2020, the only planctomycetes obtained from macroalgae were isolated in NAG medium while none were retrieved in medium M13. Lage and Bondoso also obtained a large number of *Pirellulaceae*, mostly from the genus *Rhodopirellula* and the species *R. baltica*, all from very diverse macroalgae sampled in different locations of the Portuguese coast, including Luz beach, especially from *Ulva* sp., but mainly from the thalli portions [7]. However, many of their isolates were obtained in modified medium M13. This medium has been commonly used for the isolation of planctomycetes from marine habitats, as well as the use of a supplementation with an antibiotic mixture of ampicillin and streptomycin to target planctomycetal growth, as known species are mostly resistant [7,24,38,58,59]. The problem with planctomycete isolation is the development of other fast-growing bacteria, plus the rapid and invasive growth of fungi, which can turn the purification of planctomycetes into a difficult endeavor [40]. Media 1:10 M13 and ASM are more oligotrophic and were formulated in this study in order to try to enhance the number of planctomycetal colonies obtained while decreasing the growth and number of contaminants. In fact, the strategy of using media 1:10 M13 and ASM resulted in the obtainment of significantly higher number of planctomycete strains compared to the one in M13 medium. The use of N-acetylglucosamine as the sole carbon and nitrogen source is also commonly used in planctomycetal isolation media and was thus employed in the isolation experiment in October 2020 [24,38,40]. Although only two planctomycetal strains were obtained in NAG medium, it proved to be a sufficient choice for the decrease in excessive fungal growth and bacteria from other groups, which is also consistent with the results obtained by Schlesner, who considered the utilization of media with *N*-acetylglucosamine as a carbon and nitrogen source as one of the optimal conditions for planctomycetal isolation [38]. 

### 3.2. Isolation of Planctomycetes from the Sea Water Column and Marine Invertebrates and Sediments

Only one strain closely related to *Novipirellula caenicola* was retrieved from the seawater column collected in October 2018 from Luz beach, while none were obtained from sea water collected in March or October 2020 on Memória beach (Table 3 and Figure 2). This is consistent with the knowledge that marine planctomycetes are usually less present in the water column (normally associated with marine snow) [1].

From the surface of the *Mytilus edulis* shell, we were able to isolate one strain that was 100% related to *Rubinisphaera brasiliensis* based on the analysis of the 16S rRNA gene and four strains affiliated to *Rhodopirellula baltica* in the two sampling periods, March and October 2020 (Table 3 and Figure 2). No planctomycetes were obtained from the anemone *Actinia equina* in either sampling period. Planctomycetes have already been associated to different invertebrates, such as crustaceans, jellyfish, and sponges [5,8,9,11,60,61,62]. Furthermore, in a recent study that characterized the gut microbiome of the mussel *Mytilus galloprovincialis*, [63] members of *Fuerstiella* sp., *Phycisphaera* sp., *Blastopirella* sp., and diverse unclassified planctomycetes were detected by metagenomic analysis. In fact, they are found in larger quantities inhabiting other diverse marine surfaces, such as those of macroalgae [6,7,10].

Using a standard enrichment technique for the isolation of Planctomycetes from marine sediments, we obtained five planctomycetal strains from the sand collected in March 2020, which were affiliated with the species *Novipirellula rosea* and *Gimesia chilikensis*, while no strain was obtained from the enrichment of October 2020 [24]. 

As a curiosity, we should mention that the *R. brasiliensis* strain isolated from the shell of *Mytillus edulis* and the five strains obtained from the standard sediment enrichment were isolated using the glycopeptide vancomycin to provide selection for Planctomycetes. Since Planctomycetes are Gram-negative bacteria, and glycopeptides are not capable of crossing the outer membrane of the cell wall to reach their target [64], these antibiotics proved to be an alternative for the isolation of Planctomycetes. 

### 3.3. iChip Based In-Situ Culturing System for the Isolation of Planctomycetes from Marine Sediments

Regarding the novel technique targeting the isolation of planctomycetes from marine sediments, we successfully brought three planctomycetes into pure culture (Table 3 and Figure 2). By the analysis of the 16S rRNA gene, two pink pigmented strains showed to be closely related to *Novipirellula caenicola*, while the other one, which was beige pigmented, showed to be only 96.7% affiliated to its closest relative *Rubinisphaera italica**,* based on the analysis of the 16S rRNA gene (Figure 3) [65]. This is well below the threshold for species delineation (98.7%), but above the threshold for genus delineation (94.5%), which points to a putative novel species within the genus *Rubinisphaera* [53]. With this methodology, although we also obtained additional non-planctomycetal bacterial growth in other wells, axenic cultures from the three planctomycetes were immediately obtained from their corresponding wells, which facilitated their isolation process compared to the standard methodologies. No planctomycete was isolated from the same sediment material using a standard enrichment isolation technique. Thus, this methodology based on the iChip culturing system can be an efficient approach to overcoming difficulties in the planctomycetal isolation from marine sediments, although a direct comparison of the efficiency with the traditional technique cannot be made due to the inherent differences between both methodologies The plates utilized in this experiment have a 0.22 µm filter on one side, mimicking the membranes from the original system, which have the double purpose of trapping the inoculum in the wells and preventing contamination from environmental bacteria present in the sediments; this simulates a mini diffusion chamber, which will allow for a first domestication of the bacteria and their development using only the nutrients/growth factors from their original habitat.

### 3.4. Ecology of the Isolated Planctomycetal Species

The taxa to which all the isolated planctomycetes belong have only been obtained, until present, from marine environments [3,4,6,7,41,57,66,67,68] (data from the NCBI database, August 2021). In fact, the genus *Rhodopirellula*, the genus of class *Planctomycetia* with most isolated members, is well distributed in marine environments and well associated with a variety of macroalgae, which includes diverse species found on the coast facing the Atlantic [3,4,7,10,41,69,70]. Although strains of *Rhodopirellula baltica* were found on macroalgae and in sediments and water from diverse locations (Data from the NCBI database, August 2021), we report the first isolates of this species to be associated with the biofilm of the sea mussel *Mytilus edulis* (Table 3) [7,10,41].

The *Rubinisphaera brasiliensis* type strain IFAM 1448^T^ was retrieved and described from a salt pit in Brazil, and other strains were posteriorly obtained from the post larvae of the giant tiger prawn *Penaeus monodon*, and from the macroalga *Gracilaria bursa-pastoris* from Aveiro (Portugal) (data retrieved from NCBI, August 2021) [7,62,67]. In our study, we also report its association with the biofilm of the *Mytilus edulis* shell.

*Novipirellula caenicola* type strain YM26-125^T^ was isolated from marine iron sand in Japan, but this species has also been isolated from other marine sources such as hydrothermal vent fields in the South West Pacific, sediments from Panglao Island (Philippines) (Data from NCBI database, August 2021), and the macroalgae *Fucus spiralis* and *Ulva* sp. (strains FF4 and UF2) retrieved in Foz beach near Porto (Portugal) [7,68]. *Novipirellula rosea* type strain LHWP3^T^ was retrieved from a dead ark clam belonging to species *Scapharca broughtonii*, and additional isolates were retrieved from the macroalgae *Fucus spiralis* and *Ulva* sp. (Portuguese coast) and from metalliferous deposits in hydrothermal vent fields (Southwest Pacific) (Data from NCBI database, August 2021) [7,71]. Here, we report the first isolates of this species from marine sediments.

*Gimesia chilikensis* type strain JC646^T^ was retrieved from sediments collected in the Chilika lagoon (India), but it is well distributed in marine habitats (Data from NCBI database, August 2021), including marine sediments from other locations, diverse macroalgae species, sponges, and water and metalliferous deposits in hydrothermal vent fields [5,7,72,73]. 

Regarding *Alienimonas chondri*, besides our two isolates retrieved from the macroalga *Chondrus crispus* (which include the species type strain), there is currently no data on other isolates/uncultured clones in the NCBI database with an affiliation to this species (>98.7% similarity of the 16S rRNA gene) [57]. 

## 4. Conclusions

With this study, we have enlarged the number of planctomycetes available in axenic cultures and showed that two beaches from the Portuguese north-coast have phylogenetic diversity of planctomycetes closely related to species from the families *Planctomycetaceae* and *Pirellulaceae*. The majority of the isolates were retrieved from the macroalgae communities, but we also report Planctomycetes obtained from sediments, the sea water column, and from the biofilm of the mussel *Mytilus edulis* shell. Moreover, we employed diverse medium formulations that were effectively utilized by Planctomycetes and demonstrated that more oligotrophic media were the best choices for selecting planctomycetal growth. Furthermore, we used an iChip based in-situ culturing system to successfully isolate planctomycetes from marine sediments, which included phylogenetic novelty. This technique should thus be applied to the isolation of planctomycetes from other sources such as terrestrial soil or freshwater. 

## Figures and Tables

**Figure 1 microorganisms-09-02078-f001:**
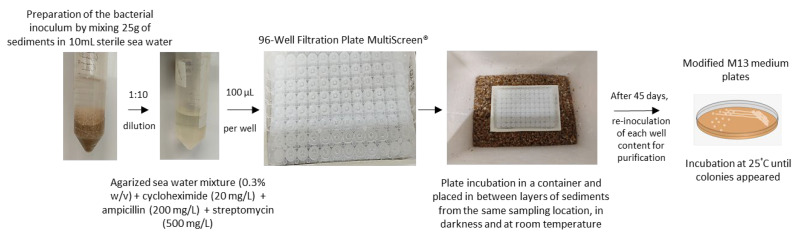
Brief description of the iChip inspired culturing system used in this study for planctomycetes isolation from marine sediments retrieved from Memória beach (Portugal).

**Figure 2 microorganisms-09-02078-f002:**
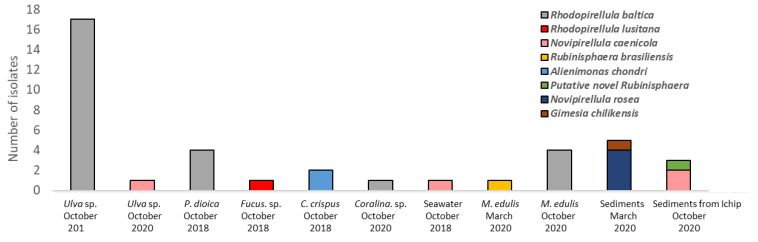
Number of isolates obtained from the different samples and collection dates. The different colors are indicative of the phylogenetic affiliation of the isolates.

**Figure 3 microorganisms-09-02078-f003:**
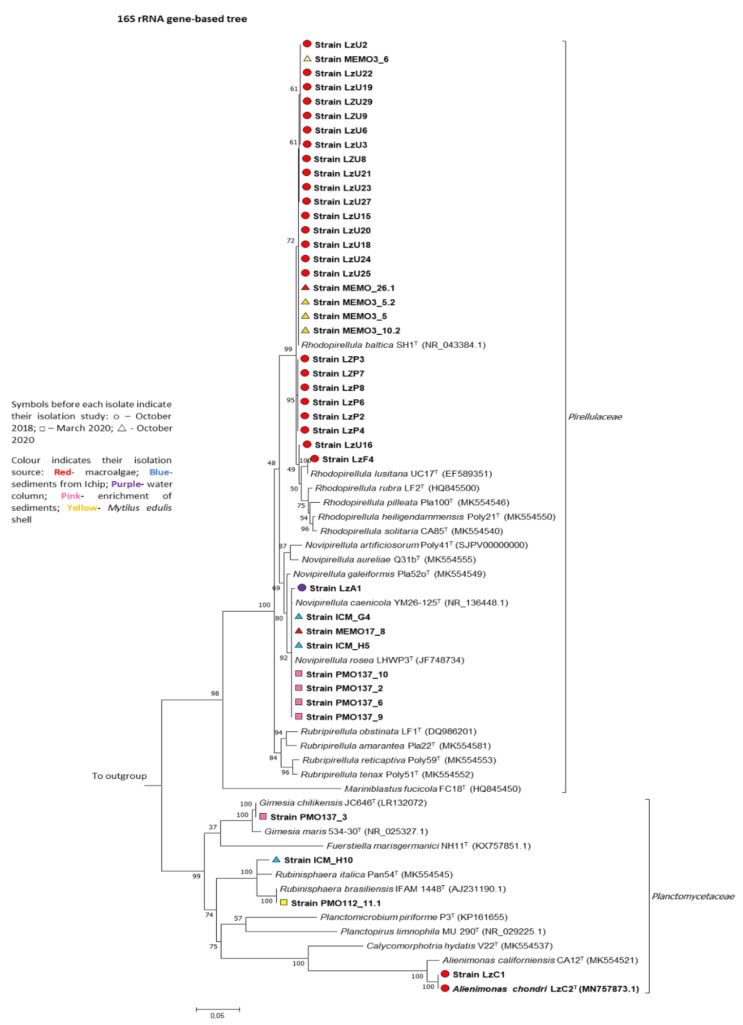
Phylogenetic 16S rRNA gene-based tree showing the affiliation of the isolates obtained in this study (in bold) within the phylum *Planctomycetes* and class *Planctomycetia*. The numbers on the branches refer to the percentage of trees in which the associated taxa clustered together from the total of 1000 bootstrap replications. The scale bar refers to 0.05 substitutions per nucleotide position. Members belonging to the Class *Phycisphaerae* were used as outgroup. Publicly available sequences from close related species type strains were obtained from the NCBI database with the respective GenBank accession numbers presented in parenthesis.

**Table 1 microorganisms-09-02078-t001:** Details of the three planctomycetal isolation experiments.

Date	Source	Media	Antibiotic Treatment	Methodology		Location
October 2018	*Chondrus crispus*	1:10 M13, M13 and ASM	Ampicillin+ streptomycin or none	Pieces of the thallus/ biofilm from the macroalgae surface	Traditional	Luz beach
	*Ulva* sp.					
	*Porphyra dioica*					
	*Fucus* sp.					
	Sea water			Water filtration (0.22 µm pore)		
March 2020	*Actinia* equina	M14 and M13+ NAG	Ampicillin+ streptomycin or vancomycin or imipenem or ciprofloxacin	Body maceration	Traditional	Memória beach
	*Mytilus edulis*			Biofilm of the shell		
	Sediments			Enrichment in liquid medium		
	Sea water			Water filtration (0.22 µm pore)		
October 2020	*Mytilus edulis*	M13 and NAGM	Ampicillin+ streptomycin	Biofilm of the shell	Traditional	Memória beach
	*Ulva* sp.			Pieces of the thallus/ biofilm from the macroalgae surface		
	*Codium* sp.					
	*Porphyra dioica*					
	*Corallina* sp.					
	Sea water			Water filtration (0.22 µm pore)		
	*Actinia equina*			Body maceration		
	Sediments	M13+ NAG		Enrichment in liquid medium		
	Sediments	Natural medium		In-situ iChip based culturing system	Novel	

**Table 2 microorganisms-09-02078-t002:** Composition of the culture media used.

Reagents	Units per Liter					
	M13 Medium ^a^	1:10 M13 Medium	Ammonium Sulfate Medium (ASM)	*N*-acetylglucosamine Medium (NAGM)	M14 Medium ^a^	M13+ NAG Medium
Peptone	0.25 g	0.025 g	-	-	1 g	0.25 g
Yeast extract	0.25 g	0.025 g	-	-	1 g	0.25 g
0.1 mM HCl-Tris buffer, pH 7.5	50 mL	50 mL	50 mL	50 mL	50 mL	50 mL
Natural sea water filtrated trough a 0.22µm filter	900 mL	909 mL	900 mL	880 mL	880 mL	900 mL
Deionized water	10 mL	10 mL	10 mL	-	-	10 mL
Glucose solution in deionized water (2.5%) ^1^	10 mL	1 mL	10 mL	-	40 mL	-
Vitamins solution ^1,2^	10 mL	10 mL	10 mL	10 mL	10 mL	10 mL
Hutner’s solution ^1,3^	20 mL	20 mL	20 mL	20 mL	20 mL	20 mL
Ammonium sulfate (NH_4_SO_4_)	-	-	10 g	-	-	-
*N*-acetylglucosamine solution in deionized water (5%) ^1^	-	-	-	40 mL	-	10 mL

^1^ These components were sterilized through a 0.22 µL pore filter and added to the medium after autoclaving; ^2^ 0.1 μg mL−1 Cyanocobalamin, 2.0 μg mL−1 biotin, 5.0 μg mL−1 thiamine-HCl, 5.0 μg mL−1 Ca-pantothenate, 2.0 μg mL−1 folic acid, 5.0 μg mL−1 riboflavin, 5.0 μg mL−1 nicotinamide; ^3^ 99 mg/L FeSO_4_.7H_2_O, 12.67 mg/L NaMoO_4_.2H_2_O, 3.34 g/L CaCl_2_.2H_2_O, 29.70 g/L MgSO_4_.7H_2_O, 50 mL/L “44” Metals, and 10.0 g/L Nitrilotriacetic acid. For 100 mL of “44” Metals: 250 mg Ethylenediaminetetraacetic acid, 1095 mg ZnSO_4_.7H_2_O, 500 mg FeSO_4_.7H_2_O; 154 mg MnSO_4_.H_2_O, 39.2 mg CuSO_4_.5H_2_O; 24.8 mg Co (NO_3_)2.6H_2_O, 17.7 mg Na_2_B_4_O_7._10H_2_O. ^a^ designations abbreviated from the modified M13 medium and modified M14 medium formulated according to Lage and Bondoso [7].

**Table 3 microorganisms-09-02078-t003:** Data on isolates obtained regarding their sampling location, date of sampling, source of isolation, medium of isolation, and affiliation.

Isolate Designation	Beach of Sampling	Date of Sampling In-Situ	Source *	Medium of Isolation **	16S rRNA Gene Similarity with the Closest Species Type Strain
LzU2	Luz	10/2018	P.*Ulv*	1:10 M13+ Pevaril® + Ant	99.4% *Rhodopirellula baltica*
LzU3	Luz	10/2018	P.*Ulv*	1:10 M13+ Pevaril® + Ant	99.8% *Rhodopirellula baltica*
LzU6	Luz	10/2018	P.*Ulv*	1:10 M13+ Pevaril® + Ant	99.8 % *Rhodopirellula baltica*
LzU8	Luz	10/2018	P.*Ulv*	1:10 M13+ Pevaril® + Ant	99.8% *Rhodopirellula baltica*
LzU9	Luz	10/2018	P.*Ulv*	1:10 M13+ Pevaril® + Ant	99.8% *Rhodopirellula baltica*
LzU15	Luz	10/2018	P.*Ulv*	1:10 M13+ Pevaril®	100.0% *Rhodopirellula baltica*
LzU16	Luz	10/2018	P.*Ulv*	1:10 M13+ Pevaril®	99.4% *Rhodopirellula baltica*
LzU18	Luz	10/2018	P.*Ulv*	1:10 M13+ Pevaril®	100.0% *Rhodopirellula baltica*
LzU19	Luz	10/2018	P.*Ulv*	1:10 M13+ Pevaril®	99.8% *Rhodopirellula baltica*
LzU20	Luz	10/2018	P.*Ulv*	ASM + Pevaril®	100.0% *Rhodopirellula baltica*
LzU21	Luz	10/2018	P.*Ulv*	ASM + Pevaril®	99.7% *Rhodopirellula baltica*
LzU22	Luz	10/2018	P.*Ulv*	ASM + Pevaril®	99.8% *Rhodopirellula baltica*
LzU23	Luz	10/2018	P.*Ulv*	ASM + Pevaril®	99.7% *Rhodopirellula baltica*
LzU24	Luz	10/2018	P.*Ulv*	ASM + Pevaril®	99.9% *Rhodopirellula baltica*
LzU25	Luz	10/2018	P.*Ulv*	ASM + Pevaril®	99.9% *Rhodopirellula baltica*
LzU27	Luz	10/2018	P.*Ulv*	ASM + Pevaril®	99.7% *Rhodopirellula baltica*
LzU29	Luz	10/2018	P.*Ulv*	ASM + Pevaril®	99.6% *Rhodopirellula baltica*
LzP3	Luz	10/2018	P. *Por*	ASM + Pevaril®	99.6% *Rhodopirellula baltica*
LzP6	Luz	10/2018	P. *Por*	1:10 M13+ Pevaril® + Ant	99.2% *Rhodopirellula baltica*
LzP7	Luz	10/2018	P. *Por*	1:10 M13+ Pevaril® + Ant	99.5% *Rhodopirellula baltica*
LzP8	Luz	10/2018	P. *Por*	1:10 M13+Pevaril® + Ant	99.2% *Rhodopirellula baltica*
LzF4	Luz	10/2018	P. *Fuc*	ASM + Pevaril®	100.0% *Rhodopirellula lusitana*
LzA1	Luz	10/2018	Sea water	M13+ Pevaril® + Ant	99.5% *Novipirellula caenicola*
LzC1	Luz	10/2018	SC.*Cho*	M13+ Pevaril® + Ant	100.0% *Alienimonas chondri*
LzC2	Luz	10/2018	SC.*Cho*	M13+ Pevaril®	100.0% *Alienimonas chondri*
PMO112_11.1	Memória	03/2020	SC.*Myt*	M14+ Pevaril® + Van	100.0% *Rubinisphaera brasiliensis*
PMO137_2	Memória	03/2020	Sediments	M13+NAG+ Pevaril® + Van	99.9% *Novipirellula rosea*
PMO137_6	Memória	03/2020	Sediments	M13+NAG+ Pevaril® + Van	99.9% *Novipirellula rosea*
PMO137_9	Memória	03/2020	Sediments	M13+NAG+ Pevaril® + Van	99.9% *Novipirellula rosea*
PMO137_10	Memória	03/2020	Sediments	M13+NAG+ Pevaril® + Van	99.9% *Novipirellula rosea*
PMO137_3	Memória	03/2020	Sediments	M13+NAG+ Pevaril® + Van	99.9% *Gimesia chilikensis*
MEMO3_6	Memória	10/2020	SC.*Myt*	M13+ cycloheximide + Ant	99.8 % *Rhodopirellula baltica*
MEMO3_5	Memória	10/2020	SC.*Myt*	M13+ cycloheximide + Ant	100.0% *Rhodopirellula baltica*
MEMO3_5.2	Memória	10/2020	SC.*Myt*	M13+ cycloheximide + Ant	99.9% *Rhodopirellula baltica*
MEMO3_10.2	Memória	10/2020	SC.*Myt*	M13+ cycloheximide + Ant	99.9% *Rhodopirellula baltica*
MEMO17_8	Memória	10/2020	P.*Ulv*	NAGM + cycloheximide + Ant	99.7% *Novipirellula caenicola*
MEMO_26.1	Memória	10/2020	Sc.*Cor*	NAGM + cycloheximide + Ant	99.9% *Rhodopirellula baltica*
ICM_H5	Memória	10/2020	Sediments from iChip	Natural medium + cycloheximide + Ant	99.7 % *Novipirellula caenicola*
ICM_G4	Memória	10/2020	Sediments from iChip	Natural medium + cycloheximide + Ant	99.7 % *Novipirellula caenicola*
ICM_H10	Memória	10/2020	Sediments from iChip	Natural medium + cycloheximide + Ant	96.7% *Rubinisphaera italica*

* P.*Ulv*: Portions of *Ulva* sp.; P.*por*: portions of *Porphyra dioica*; P.*Fuc*: portions of *Fucus* sp.; SC.*Cho*: scrapped biofilm of *Chondrus crispus*; SC.*Myt*: scrapped biolfim of *Mytilus* sp. shell; SC.*Cor*: scrapped biofilm of *Corallina* sp. **; Ant: antibiotic supplementation with streptomycin and ampicillin; Van: antibiotic supplementation with vancomycin; Pevaril®: Econazole nitrate.

## Data Availability

Not applicable.

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
