# Peer review of "Novel and Conventional Isolation Techniques to Obtain Planctomycetes from Marine Environments"

_microorganisms, 2021, doi:10.3390/microorganisms9102078_

Round 1

Reviewer 1 Report

The article by Inês Vitorino et al., “Novel and conventional isolation techniques to obtain Planctomycetes from marine environments” reports the isolation of 40 marine members of the Planctomycetes phylum from the Portuguese north coast using different conventional and novel techniques. Thanks for your work increasing the number of strains available as axenic cultures.

The manuscript is well written, however I have some minor issues.

Minor comments:

Line 39:  the phylum Planctomycetes is also containing the Candidatus class Brocadiae.

Line 97: replace “planctomycete” by “planctomycetes”

Line 122: replace “five hundred” by “500”

Major comments:

- Why the enrichment of the second isolation was incubated during one year and the enrichment of the third isolation only 45 days?

- Figure 2 is not self-explicative. Looking at the figure seems like the isolates are the one described directly in the legend but they are supposed to be related to them. Please explain better in the figure legend.

- If I understood correctly the iChip isolation method was validated comparing with the results obtained using  the classic enrichment technique however the conditions were not the same. The inoculum for the iChip was done in distillated water while for the classical enrichment M13+NAG was used. The volume (of the inoculum, 10 vs 25 ml), temperature (20 vs 25 ºC) and agitation were not the same in both assays. Also it is discussed that the amount of isolates is higher when the media are more oligotrophic however in this assays you are comparing with M13+NAG which is richer.

- In the iChip assay you plated one plate for each well, so 96 plates. However from the enrichment you only plated 100 ul. I think is difficult to compare the success of the technique if you don’t plate the same proportional amount. In case you did, please clarify it.

- Regarding the 40 isolates I am not sure if all of them are novel since the similarity of them with their related species is near 100%. Only one strain, the isolate ICM_H10, is below the threshold for species.

- Line 331-333: it is said that this system could prevent contamination from environment. I see this point when the wells are inoculated with one  single cell coming from FACS however when they are inoculated with material coming from the sediments they already containing everything that was there.

Author Response

The authors are very grateful to the reviewer for all the comments and suggestions. Please find bellow the answers to all the points raised.

Minor comments:

 Line 39:  the phylum Planctomycetes is also containing the Candidatus class Brocadiae.

R: The following sentence was added: “Additionally, the Candidatus order "Brocadiales" from the Candidatus class “Brocadiae” is constituted by the anaerobic ammonium oxidation (anammox) planctomycetes, which do not have axenic cultures available yet”.

Line 97: replace “planctomycete” by “planctomycetes”

R: This point was corrected in the manuscript.

Line 122: replace “five hundred” by “500”

R: This point was corrected in the manuscript.

Major comments:

 - Why the enrichment of the second isolation was incubated during one year and the enrichment of the third isolation only 45 days?

R: This, unfortunately, happen due to the pandemic situation, as access to the laboratory has been restricted during a long period of time.

- Figure 2 is not self-explicative. Looking at the figure seems like the isolates are the one described directly in the legend but they are supposed to be related to them. Please explain better in the figure legend.

R: The legend of the figure was changed to: “Number of isolates obtained from the different samples and collection dates. The different colours are indicative of the phylogenetic affiliation of the isolates”.

- If I understood correctly the iChip isolation method was validated comparing with the results obtained using the classic enrichment technique however the conditions were not the same. The inoculum for the iChip was done in distillated water while for the classical enrichment M13+NAG was used. The volume (of the inoculum, 10 vs 25 ml), temperature (20 vs 25 ºC) and agitation were not the same in both assays. Also it is discussed that the amount of isolates is higher when the media are more oligotrophic however in this assays you are comparing with M13+NAG which is richer.

R: The iChip was not validated by the traditional isolation method. It was used as an alternative methodology. We must note that the iChip was incubated in sea water and not in distilled water.

- In the iChip assay you plated one plate for each well, so 96 plates. However from the enrichment you only plated 100 ul. I think is difficult to compare the success of the technique if you don’t plate the same proportional amount. In case you did, please clarify it.

R: The following sentence was added: “No planctomycete was isolated from the same sediment material using a standard enrichment isolation technique. This methodology based on the iChip culturing system can, thus, be a good approach to overcome difficulties in planctomycetal isolation from marine sediments, although a direct comparison of the efficiency with the traditional technique cannot be made due to the inherent differences between both methodologies.”.

- Regarding the 40 isolates I am not sure if all of them are novel since the similarity of them with their related species is near 100%. Only one strain, the isolate ICM_H10, is below the threshold for species.

 R: These are all new isolates but with close phylogenetic similarity to described species with the exception of strain ICM_H10 which is a new non yet described species.  

- Line 331-333: it is said that this system could prevent contamination from environment. I see this point when the wells are inoculated with one single cell coming from FACS however when they are inoculated with material coming from the sediments they already containing everything that was there.

R: What we meant is that, when the plate is closed, no more microorganisms can “invade” the wells. The ones that are going to grow are the ones that were inoculated before sealing the plate. With the performed dilution, it is expected very low number, ideally one, per well. The 0.22µm membrane avoid the undesirable contaminations from the environment.

Reviewer 2 Report

The manuscript has well been designed, conducted, described and analyzed. The authors have aimed to isolate and identify specias of Plactomycetes growing in different habitats, using a conventional technique based on cultivation. However, they have also modified this technique to obtain more isolates from the samples.

Author Response

The authors are very grateful to the reviewer for the acceptance of the paper.